# Calculating curly arrows from ab initio wavefunctions

Yu Liu[1], Philip Kilby[2], Terry J. Frankcombe[3] & Timothy W. Schmidt [1]

Despite being at the heart of chemical thought, the curly arrow notation of reaction mechanisms has been treated with suspicion—the connection with rigorous molecular quantum mechanics being unclear. The connection requires a view of the wavefunction that goes beyond molecular orbitals and rests on the most fundamental property of electrons. The antisymmetry of electronic wavefunctions requires that an $N$-electron wavefunction repeat itself in $3N$ dimensions, thus exhibiting tiles. Inspection of wavefunction tiles permits insight into structure and mechanism. Here, we demonstrate that analysis of the wavefunction tile along a reaction coordinate reveals the electron movements depicted by the curly arrow notation for several reactions. The Diels–Alder reaction is revealed to involve the separation and counter propagation of electron spins. This unprecedented method of extracting the movements of electrons during a chemical reaction is a breakthrough in connecting traditional depictions of chemical mechanism with state-of-the-art quantum chemical calculations.

[1] ARC Centre of Excellence in Exciton Science, School of Chemistry UNSW Sydney Sydney, NSW 2052, Australia. [2] Data 61, Locked Bag 8001, Canberra, ACT 2601, Australia. [3] School of Physical Environmental and Mathematical Sciences, UNSW Canberra, Canberra, ACT 2600, Australia. Correspondence and requests for materials should be addressed to T.W.S. (email: timothy.schmidt@unsw.edu.au)

The 'curly arrow' notation for depicting the mechanisms of chemical reactions was devised by Robinson in the early 1920s[1,2], and was further developed by others, including Ingold[3,4], who established the double-headed arrow for the displacement of a pair of electrons. The single-headed, 'fish-hook' notation for a single electron was introduced in 1965 by Budzikiewicz, Djerassi and Williams[5], completing the basis of how organic reaction mechanisms have been depicted to millions of students and in myriad research papers since. Nevertheless, the concept of the curly arrow notation continues to be developed[6-8].

Curly arrows are a convenient electron accounting tool, and for many that is all they are. Indeed, the curly arrow notation relies on being able to draw Lewis structures, whereby electrons are found localised as bonds, or 'lone pairs'[9]. However, this is at odds with the spirit of molecular orbital (MO) theory which gives no consideration to explicit bond descriptions without post-processing of the calculated wavefunction. Indeed, raw MO theory tends to delocalise electrons in 'orbitals' which may range over an entire molecule.

At the heart of MO theory is the Hartree–Fock method for calculating electronic structure, in which each electron is held to move in the averaged potential of the other electrons (including terms to account for exchange). The effective one-electron Hamiltonian is solved in a self-consistent manner and the resultant one-electron eigenfunctions are taken as the canonical MOs. A wavefunction calculated in this way is said to be 'single determinant', and includes no information regarding the electrostatic correlation of electrons. Electron correlation must be recovered by including 'excited' configurations, the resulting wavefunctions being denoted multi-configurational.

There are many methods to subsequently localise these orbitals, and the so-called intrinsic bonding orbitals (IBO) and electron localisation function have been used successfully to calculate electron flow in certain reaction mechanisms[10,11]. Indeed, the electron flow in many reactions well described by a single determinant was successfully described using an IBO method by Knizia and Klein[10]. However, it is well known that a single determinant wavefunction fails to describe the simplest of chemical reactions, the homolytic bond cleavage. Many molecules and transition structures simply cannot be described by a single determinant wavefunction, and thus a single set of localised orbitals. No attempts to utilise localisation schemes based on the reduced density matrix for multi-determinant wavefunctions to track the evolution of Lewis structures along a reaction path have been published in the literature.

Recently, we and others described an intuition-free method to recover electron localisation from an arbitrary wavefunction[12-15]. Our method, Dynamic Voronoi Metropolis Sampling (DVMS), relies on the indistinguishability of electrons, as well as the antisymmetry property common to all electronic wavefunctions, whereby the sign of the wavefunction changes upon spatial permutation of like-spin electrons. These properties are used to find 'wavefunction tiles', hyper-regions which repeat throughout the $3N$-dimensional $N$-electron wavefunction. Since the tile repeats, much like the unit cell of a crystal structure, only one tile is needed to represent the electronic structure.

The wavefunction tile reproduces traditional bonding motifs in simple molecules such as water, $N_2$, $O_2$, $H_2CO$ and $C_2H_4$. The qualitative nature of the tiles, of systems tested so-far, have been found to be insensitive to the size of the basis set beyond 6-31G (d)[15]. However, some inherently multi-configurational wavefunctions require a number of configurations before qualitative convergence. For example $C_2$, which is pathologically multi-configurational, was found to exhibit a distorted triple-bond with a further pair of spatially disparate singlet-coupled electrons, as demonstrated by Shaik and co-workers[15,16].

In this work, we demonstrate the following of wavefunction tiles as a function of a reaction coordinate, and that the resulting movements of electrons are a calculable representation of the 'curly arrow' notation of chemical mechanism. We demonstrate that our method succeeds even where single determinant wavefunctions fail, and that the [4+2] Diels–Alder reaction is best described as counter-propagating $\alpha$ and $\beta$ spins. Connecting rigorous quantum chemical calculations with the traditional depiction of reaction mechanism offers an exciting opportunity for insight into chemical reactions.

## Results

**Wavefunction tiles**. The wavefunction tiles are a set of non-overlapping, single-signed (for ground state) functions which regenerate the full wavefunction $\Psi(\mathbf{x})$ under the set of permutations, $\{P_j\}$, of like-spin electrons, $\Psi(\mathbf{x}) = \sum_j \psi_j(\mathbf{x})$, where $\psi_j(P_j \circ \mathbf{x}) = \pm\psi_0(\mathbf{x})$ with the sign depending on the number of single element swaps inherent in permutation $j$.

These are non-unique, as is the unit cell of any crystal. But, just as it makes sense to place the boundaries of the unit cell of a crystal on atomic planes, so too boundaries between positive and negative tiles should be delineated by a node. Boundaries between two adjoining tiles of the same sign are less clear. In DVMS[15], we chose a self-consistent method whereby a cloud of random-walkers $\{\mathbf{x}_i\}$ samples the wavefunction, $\Psi$, moving with a probability given by $\Psi^2$ (Metropolis sampling). Each walker $\{\mathbf{x}_i\}$ must remain closer to the average walker position $\bar{\mathbf{x}}$ than to any same-sign permutation of it, $P \circ \bar{\mathbf{x}}$ where $\Psi(P \circ \bar{\mathbf{x}}) = \Psi(\bar{\mathbf{x}})$. This makes the same-sign tile boundaries the boundaries of a Voronoi tesselation of the set of permuted average positions, $\{P \circ \bar{\mathbf{x}}\}$. We thus call the tile-defining average walker position, $\bar{\mathbf{x}}$, a "Voronoi site".

It was found that the average walker positions, and thus the tile boundaries, converged after about 1000 steps for the cases considered so far[15]. The convergence of each Monte-Carlo simulation was judged by the observed variation of the position of the Voronoi sites. The certainty of these positions was gauged by the error of the mean of a series of independent Monte-Carlo simulations. Since the walkers remain within the boundaries of the tile, they now sample $\psi_0(\mathbf{x})$, and seek $\bar{\mathbf{x}} = \int \mathbf{x}\psi_0^2 d\mathbf{x} / \int \psi_0^2 d\mathbf{x}$.

Visualizing the wavefunction tile, a $3N$-dimensional function, can be done a number of ways. One simple way is to project the $3N$-dimensional vector $\bar{\mathbf{x}}$ onto the set of 3-dimensional vectors $\{\bar{\mathbf{r}}_p\}$ representing the average position of each electron within the tile $\psi_0$. Structures obtained in this way are given on the left hand side of Fig. 1. Here, the electron sites are illustrated with gold spheres for water, ethylene and $N_2$, where nuclear positions are indicated with larger spheres coloured by the CPK convention.

A representation which is more recognisable to the modern, practicing chemist is an isosurface resembling an orbital. These are plotted for each electron, in turn, giving the 3-dimensional surfaces defined by $\mathcal{S}_p = \{\mathbf{r}_p | \psi_0(\bar{\mathbf{r}}_1, \bar{\mathbf{r}}_2, \dots, \mathbf{r}_p, \dots, \bar{\mathbf{r}}_N) = \zeta\}$ for the $p$-th electron. The value of $\zeta$ is chosen aesthetically. These surfaces are plotted in Fig. 1, for each of the molecules shown. Chemical motifs such as lone-pairs, single bonds and multiple 'banana bonds' ($\tau$-bonds) are naturally exhibited by the wavefunction tile. The core electrons, which appear as small spheres, are not shown.

There are some structures, such as the methyl radical, where the wavefunction tile is 'two-humped'. In such cases, the wavefunction tile should not be described by an average position, nor an isosurface of a cross-section through the average position as above. Rather, the tile should be projected onto the coordinates of each electron, $\phi_p(\mathbf{r}_p) = \int \psi_0 d\mathbf{y}$, where $\mathbf{y} = \{\mathbf{r}_{q\neq p}\}$.

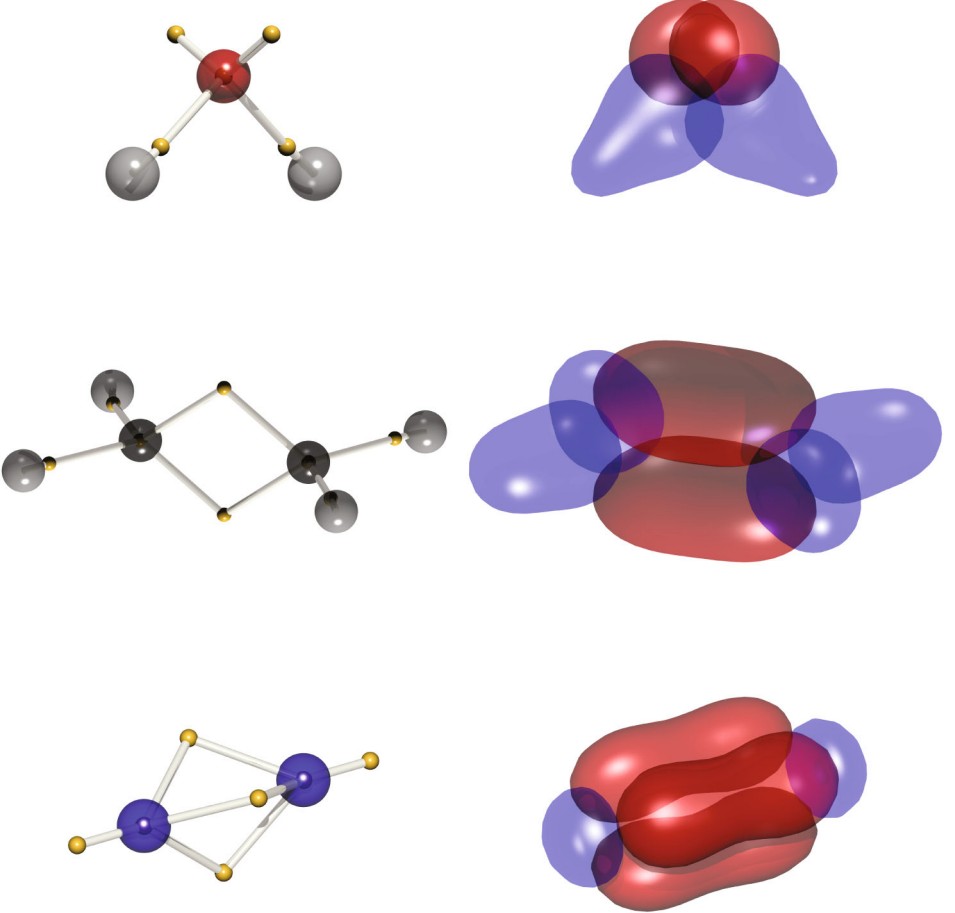

**Fig. 1** Representations of electronic structure calculated by DVMS. (Left) The RHF/6-31G(d) DVMS structures of (from top) water, ethylene and dinitrogen. The gold spheres represent the average positions of electrons in the wavefunction tile. Atomic positions are given as spheres coloured according to the CPK convention. Silver rods are drawn to join nuclear positions and electron positions. (Right) Isosurfaces for each valence electron, where the positions of all other electrons are held fixed at the positions indicated on the left. Blue and red colours are used to differentiate valence electrons of each type within a given molecule

It may be computationally convenient to break the tile down into a number of non-overlapping sub-tiles, $\psi_0 = \sum_m \phi_0^m$, in which case the projection can be made from the combined tile or an individual sub-tile, depending on the property one wants to illustrate. An automated method to determine when to split into sub-tiles can be developed through a clustering analysis. The expectation maximisation algorithm[17] can be adapted to identify whether the cloud of DVMS walkers exhibits significant bimodal character. The expectation maximisation algorithm as a cluster analysis method is well suited to identifying the overlapping clusters that would be expected to occur as wavefunction tiles bifurcate into structures well represented by multiple sub-tiles.

The projection of the DVMS random walker representation of the wavefunction tile onto the coordinates of the unpaired electron in methyl radical is shown in Fig. 2a, with the two sub-tile Voronoi sites indicated in green. Separate isosurfaces generated by cross sections through these sites are combined in Fig. 2b. The combined surface strongly resembles the unhybridized p-orbital of the chemist's intuition even though the isosurface is a slice through the $3N$-electron wavefunction tile.

With the localisation of electrons by the wavefunction tile now established, we are in a position to investigate the evolution of the tile as a function of the reaction coordinate[18]. We first demonstrate below that the $S_N2$ reaction and nucleophilic addition to a carbonyl can be followed by a representation of the wavefunction tile. Then we show two examples of the

wavefunction tile splitting along the reaction coordinate: homolytic bond cleavage and the Diels–Alder reaction. We then track the path of the Voronoi site along the reaction coordinate to retrieve a calculated electron trajectory, or curly arrow.

**$S_N2$ reaction**. Figure 3 shows wavefunction tile cross-section isosurfaces plotted along the intrinsic reaction coordinate[18] for the reaction $FCH_3 + OH^- \rightarrow F^- + CH_3OH$, calculated at the RHF/6-31G(d) level. The mechanism elucidated from analysis of the tile evolution is precisely that found in undergraduate textbooks: a lone-pair of electrons on the attacking anion form the new single bond, and both the electrons in the breaking bond leave as a lone pair on the leaving group[19].

While we stress that the wavefunction tile analysis is completely general and, in principle, does not necessitate the invocation of MOs, the localised orbital analysis of Knizia and Klein agrees with the present analysis of the $S_N2$ reaction[10]. The present analysis, being of a single-determinant wavefunction, cannot test the assertion of Shaik, that the $S_N2$ reaction proceeds first with a one-electron movement, as if to generate radical products, with the second electron catching up, the argument being that the reactant state diabatically correlates with charge-transfer products[20]. To test this assertion requires the analysis of a multiconfigurational wavefunction which is demonstrated below for the Diels–Alder reaction.

**Nucleophilic carbonyl addition**. Due to the electronegativity of oxygen, the carbon in a carbonyl group carries a partial positive charge, rendering it susceptible to attack by nucleophiles. The accepted mechanism is that the nucleophile (Nu:) donates a pair of electrons to form a nascent Nu–C bond, and one pair of electrons from the carbonyl double bond form a new lone pair on the oxygen which now carries a formal negative charge. This intuition is born out by the analysis of the wavefunction tile, as shown in Fig. 4. In Fig. 4, the structure of the reactants, formaldehyde and hydroxide, exhibits electrons localised in lone pairs, single bonds, or as one component of a banana-bond

($\tau$-bond). As the reaction proceeds, one lone pair from the hydroxide morphs into a single bond (light green), as expected, while the distal component of the $\tau$-bond morphs into a lone-pair (bottle green), the proximal component (grey) sliding into the region between the C and O nuclei to form a single bond. All other electrons (blue) remain unaltered chemically, in complete accord with intuition.

**Homolytic bond cleavage**. It is well known that a single determinant wavefunction, such as that produced by the Hartree–Fock (HF) method, cannot describe homolytic bond cleavage, even for $H_2$. Coulson and Fischer showed this to be due to over-emphasis of the ionic terms at large internuclear distance and proposed configuration interaction[21], thus reconciling the MO treatment with the Heitler-London wavefunction[22], which is the father of valence bond theory. Indeed, Coulson and Fischer warned of 'the dangers inherent in too naive an application of MO theory to interactions across large distances, as, for example, in trying to follow the course of a chemical reaction'.

The results of a 2-configuration calculation of the HF→H+F bond cleavage are shown in Fig. 5. The positions of walkers in the wavefunction tile, projected onto the $z$-coordinates of each of the bonding electrons, is shown as a function of the internuclear distance. At equilibrium (0.91 Å), this distribution is clearly monomodal, and the tile is well represented by a single Voronoi site. However, upon stretching of the bond, the tile evidently bifurcates into two regions, each corresponding to either the $\alpha$ or $\beta$ electron of the bond becoming associated with the hydrogen atom. From a bond length of 1.11 Å, the effects of configuration interaction on the wavefunction are clear, as the $z$-distribution of each of these electrons is correlated with the other.

As in the methyl radical case above, this necessitates a description of the wavefunction tile with two Voronoi sites, yielding two sub-tiles. Projecting only one of the sub-tiles onto the coordinates of each electron, Fig. 5 shows the blue-coloured electron dissociating from the F atom, while the red-coloured electron remains associated with it. Plotting the other sub-tile simply has this colour scheme reversed.

As such, plotting one sub-tile of the wavefunction allows a pictorial description of the effects of electron correlation in homolytic bond cleavage, a process which a single determinant wavefunction cannot properly describe. This splitting of the wavefunction is also exhibited by the Diels–Alder reaction.

**[4+2] Diels–Alder reaction**. The explanation of the reactivity of the Diels–Alder reaction is a triumph of MO theory. As explained by the Woodward–Hoffman rules[23], the symmetries of the frontier orbitals dictate whether a cyclization reaction will occur

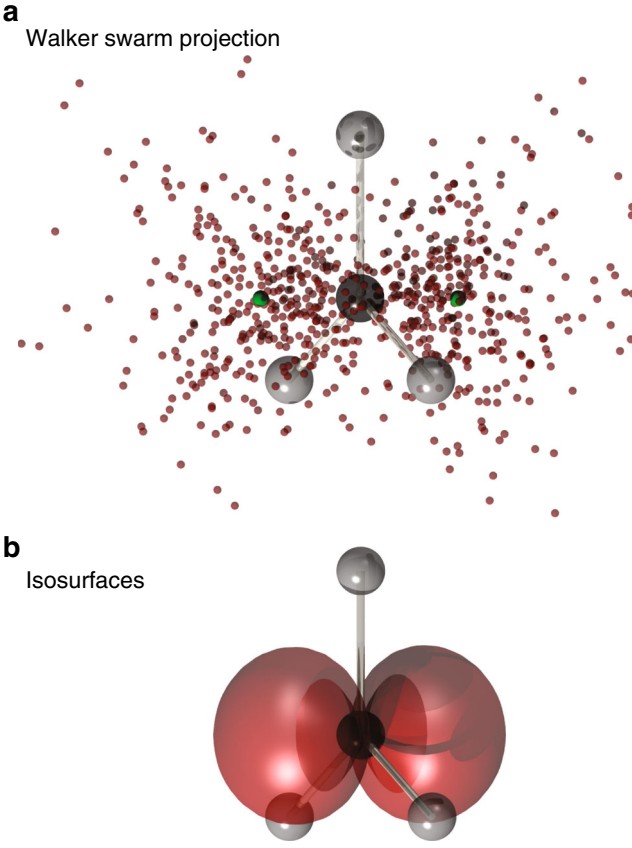

**a** Walker swarm projection

**b** Isosurfaces

**Fig. 2** Projections of the wavefunction tile for the methyl radical. **a** The projection of the walkers representing the wavefunction tile onto the coordinates of the unpaired electron of methyl radical. The green spheres represent the Voronoi sites of the two sub-tiles. **b** The isosurfaces generated by cross sections through the Voronoi sites in **a**

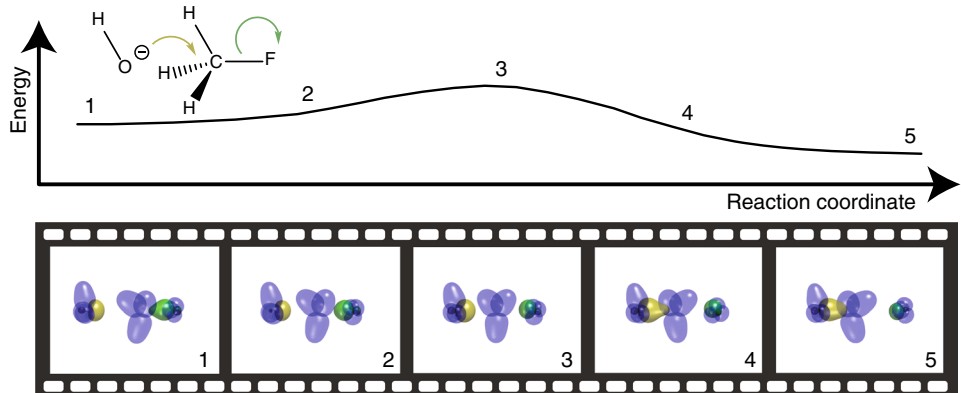

**Fig. 3** Wavefunction tile cross-section isosurfaces of the progress of the $S_N2$ Reaction of $FCH_3+OH^-\rightarrow F^-+CH_3OH$

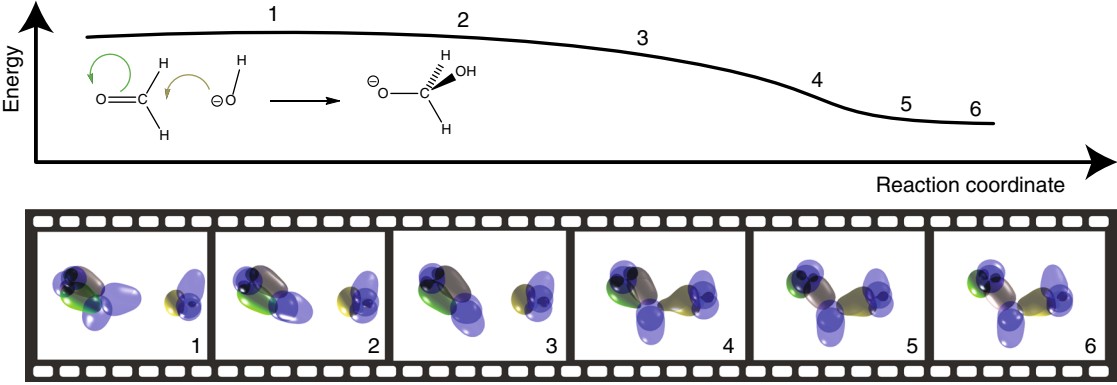

**Fig. 4** The nucleophilic addition of hydroxide anion to a carbonyl group. Both electron spins follow the same path

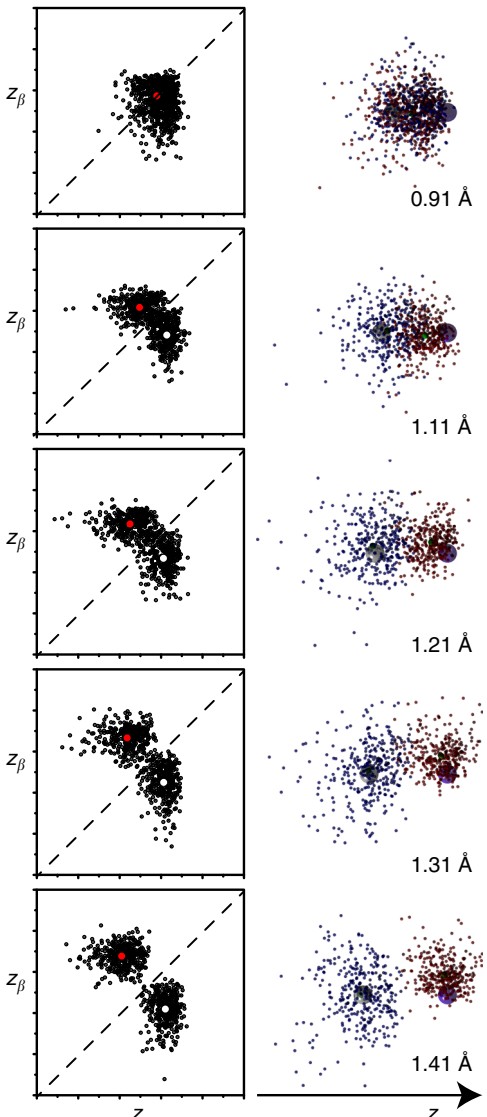

**Fig. 5** The homolytic bond cleavage of hydrogen fluoride to yield fluorine and hydrogen atoms. For bond lengths 1.11 Å and longer, the wavefunction is two-humped, necessitating two Voronoi points (red and white). Projection of one sub-tile yields the structures plotted on the right

thermally or photochemically. The concerted mechanism[24] is usually drawn showing the cyclic motion of three electron pairs (Fig. 6a, inset)[25], notwithstanding disclaimers regarding the actual mechanism[26]. However, spin-coupled valence bond theory calculations suggest a homolytic mechanism, drawn with six half-arrows (Fig. 6b, inset)[27]. Many organic chemistry textbooks decline to offer a curly arrow mechanism[19], and the MO theory interpretation suggests a smooth evolution between delocalised orbitals in the reactants and the products. The wavefunction tiles tell an intriguing story.

Figure 6 illustrates the wavefunction tile as a function of the intrinsic reaction coordinate for the cycloaddition of butadiene and ethylene. C–H bonds are coloured grey and are essentially unchanged during the reaction. Lobes that stay bonding between a particular pair of atoms are coloured aqua, and those poised to homolytically split are coloured purple in the initial structure. The transition state is very reactant-like. Immediately, on the product side of the transition state, the wavefunction tile bifurcates into a form requiring a description in terms of two sub-tiles, and only one sub-tile is shown (the other is the form with $\alpha$ and $\beta$ electrons, respectively, moving in the other sense). The banana-bond on the butadiene side of the ethylene molecule splits, with each component heading to form half of a new C–C bond (the two spins are coloured blue and red). Meanwhile, the ethylene-ward banana bonds of butadiene also split, with one of each component completing the new C–C bonds and the others migrating into the central C–C bond of butadiene to form a new banana-bond. The existing C–C bond is pushed aside, making up the second component of the new double (banana) bond. This reaction occurs on the singlet surface, and despite the unpairing of electrons during the reaction, the singlet and triplet states are not near-degenerate.

Certainly, the evolution of the wavefunction tile is strongly suggestive of the homolytic mechanism as depicted in Fig. 6b. In what follows, we calculate the curly arrow path by plotting the Voronoi site of the wavefunction sub-tile as function of the reaction coordinate, and show that this is indeed the case.

**Calculating the curly arrow path**. The Voronoi site of the wavefunction (sub-)tile is a convenient proxy for electron location. It is rigorously the average position of the electrons in the (sub-)tile, and we have shown previously that these locations exactly give the dipole moment of a molecule and a good approximation to the quadrupole moment[15]. The locus of the Voronoi site as a function of reaction coordinate thus links the 'electron positions' along the reaction coordinate. Notwithstanding Monte-Carlo noise, these are smooth lines.

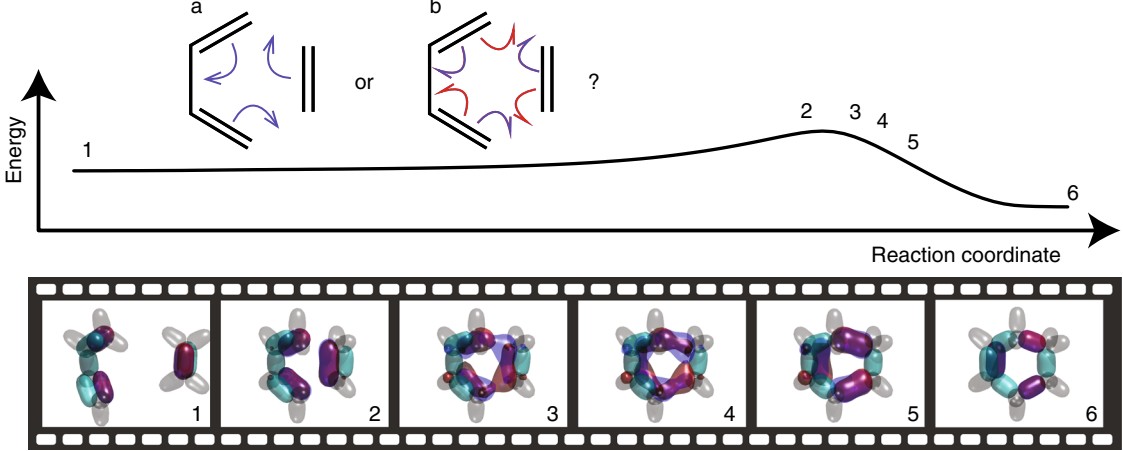

**Fig. 6** The evolution of one wavefunction sub-tile along the Diels–Alder reaction coordinate. While frequently drawn as concerted electron pair motion (inset a), DVMS shows electrons of different spin counter-propagate (inset b), supporting the homolytic mechanism

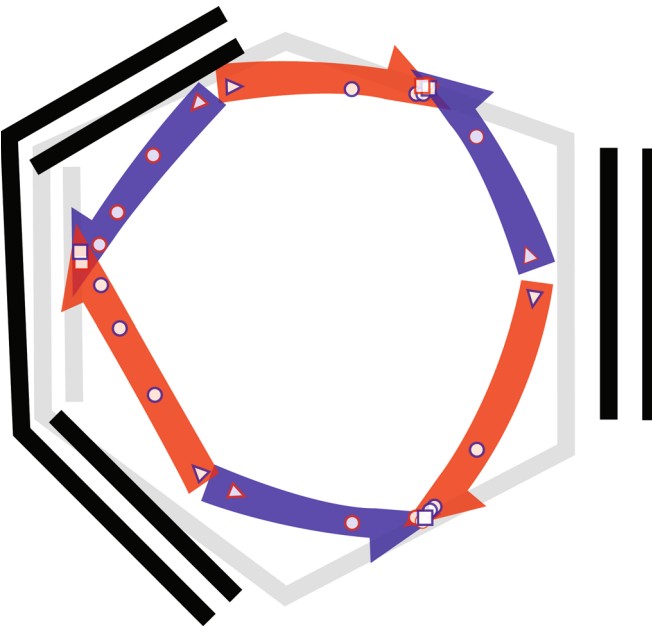

**Fig. 7** The calculated curly arrow mechanism of the Diels–Alder reaction of 1,3-butadiene and ethylene. The apparently distorted geometries arises from the 3-dimensional reactant (black) and product (grey) geometries being projected onto the plane. The curved paths of the electrons are drawn through the calculated intermediate structures (start: triangle, end: square)

In Fig. 7, the initial and final geometries in the [4+2] Diels–Alder reaction are drawn in black and grey. They are a projection of 3-dimensional geometries onto the plane, and thus appear slightly distorted. The electron loci are plotted at five positions along the reaction coordinate (corresponding to frames 2–6 in Fig. 6), with the starting position as a triangle, and the final position as a square. Single-headed arrows indicate the path of each electron, with colours denoting opposing spins.

The wavefunction tile interpretation clearly supports the homolytic mechanism put forward by valence-bond calculations. Furthermore, recovering this from MO calculations requires multiple configurations to correlate the electrons, and thus this result could not be obtained from a single-determinant treatment, or by simply plotting orbitals as a function of the reaction coordinate. While our results for the Diels–Alder reaction support a homolytic mechanism, we do not propose that this necessarily applies to the similarly pericyclic Claisen rearrangement which was found by Knizia and Klein to undergo a cyclic movement of three electron pairs[10], as their method is bound to conclude.

## Discussion

We have demonstrated that the analysis of any electronic wavefunction in terms of the tiling due to antisymmetry yields a picture of electronic structure in which the electrons are naturally localised. The wavefunction tiles may be viewed as a function of a reaction coordinate to ascertain the electron movements during a chemical reaction. Some reactions involve a bifurcation of the wavefunction tile, necessitating sub-tiles to describe the wavefunction.

Plotting the locus of the Voronoi site (centroid) of a wavefunction tile reveals the 'curly arrow' beloved of students of organic reaction mechanisms. We have shown this method to be completely general, the curly arrow mechanism of any chemical reaction can be calculated this way. It is applicable to single-determinant and multi-determinant wavefunctions alike—the method only relies on there being a wavefunction which obeys Fermi–Dirac statistics.

To the best of our knowledge, this is the first time that a general electronic wavefunction has been connected to the curly arrow notation, representing a significant step forward in the rationalisation of chemical thought.

## Methods

All wavefunctions were calculated using GAMESS package[28], using the 6-31G(d) basis set[29]. Hartree–Fock orbitals were used to construct a reference configuration, and 'excited' configurations. The number of configurations used for each wavefunction were as follows: HF (2), Diels–Alder (3), $S_N2$ (1), nucleophilic addition (2). Geometries along the intrinsic reaction coordinate were generated using Gaussian 09[18,30].

Wavefunction tiles were interrogated using the DVMS procedure outlined in our previous work[15]. Permutations decomposable as an even number of element swaps were computed using a custom algorithm in which the computational effort scales as $\mathcal{O}(N \log N)$. This algorithm is not yet adapted for the permutations required for a multiple Voronoi site DVMS analysis, so a primitive factorial-scaling algorithm was used in these cases.

**Computer code availability**. The computer code used to support the findings of this study are available from the corresponding author upon reasonable request.

**Data availability**. The data that support the findings of this study are available from the corresponding author upon reasonable request.

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

## Acknowledgements

T.W.S. acknowledges the Australian Research Council for a Future Fellowship (FT130100177). This work was supported by the Australian Research Council Centre of Excellence in Exciton Science (CE170100026). Y.L. is the recipient of an Australian Postgraduate Award.

## Author contributions

Y.L. wrote computer code, performed the calculations and prepared figures. P.K. designed the custom assignment problem algorithm. T.W.S. conceived the DVMS procedure and its application to the curly arrow problem. T.J.F. and T.W.S. wrote the manuscript.

## Additional information

**Competing interests:** The authors declare no competing interests.

