## [Peer Review File · Nature Communications]

Reviewer #1 (Remarks to the Author):

In the manuscript, the authors demonstrate that their recently developed Voronoi tiling method is capable of constructing "curly arrow" reaction mechanisms from ab initio wave functions. The physical basis of the widely used "curly arrow" description of reactions has been unclear and disputed. The method of the authors shows convincingly for a simple SN2 reaction, but also for a 4+2 Diels Alder reaction, that movements of electron distributions, resulting in the "curly arrow" description, can be constructed from a Voronoi tiling based on a $|\Psi|^2$ distribution obtained with Metropolis sampling. This work is of general interest for the chemical scientific community. Although the method behind this work was published by the authors a year ago, the application of the method to reaction paths rather than molecules is original. It is particularly interesting that the author's method does not rely on the MO model.

The paper should be published in Nature after some modifications.

(1) The authors write (p.2 2nd paragraph): [the curly arrow notation...] is at odds with MO theory which tends to delocalise electrons in orbitals. This is not quite correct. It is well known that orbitals are not unique and unitary transformations can provide localised MOs (LMO) yielding the same $|\Psi|^2$ and energy as the canonical MOs. LMOs are known to generally yield Lewis structures and thus also "curly arrows" along reaction paths. The authors should discuss similarities and differences of their method and LMO methods such as Knizia and Klein's (ref [10]).

(2) On p. 8, the authors refer to Knizia and Klein's recent paper (the reference [10] is missing), writing that Knizia's localized orbital analysis agrees with the author's results. The advantage of the new method over the well established orbital analysis remains then unclear (except for the easy use of multideterminant wave functions). It seems that Knizia's results do not fully agree with the author's results when comparing the Claisen reaction in Knizia's paper and the similar Diels-Alder reaction in the manuscript. The discussion here should be clarified.

(3) The authors sometimes use "sub-tiles" demonstrated in Fig. 2. It should be clarified what criteria are used to split a tile into sub-tiles.

Reviewer #2 (Remarks to the Author):

This is a curious manuscript that uses largely unknown (but interesting) ideas to analyse molecular wave functions. My conclusion is that the authors claim that no other methods can substantiate/illustrate "curly-arrow" (arrow-pushing) interpretations of mechanisms is flawed. One may utilise Natural Bond Orbitals to do this, as is well-known. My feeling is that this flaw suggests that this otherwise interesting work should be sent to another journal (perhaps JACS?) rather than to Nature.

Reviewer #3 (Remarks to the Author):

This manuscript describes the application of a very interesting "tile-based" analysis of multi electron wave functions to examine localized electron densities and the movement of densities during chemical reactions. The DVMS method has been described separately and is not the focus of the manuscript. Rather the focus of this manuscript is the application of this method to the analysis of electron density flow during chemical reactions for both single determinant and multi-determinant wavefunctions. Although described reasonably clearly, the way the method "works" is beyond the general readership but the way the method "works out" is relatively clear. Overall, this is a novel and intriguing addition to the literature for which publication is warranted, providing that some clarity is provided on the following points.

In the analysis of wave functions and densities, uniqueness of a particular representation is a desirable property. It would be helpful if the authors explicitly addressed the following points: (1) what is the criterion for convergence of the DVMS representations (2) are the qualitative DVMS

representations sensitive to the basis set size? (3) For multideterminantal wave functions, does the representation exhibit high sensitivity to, say, the size of the active space? (4) How does one determine when a representation requires two or more sub-tiles?

To me, it appears that the properties of other bond localization methods are mischaracterized. For example, density-based methods (ALMO, NBO, NOCV, etc) can certainly analyze the alpha and beta one-electron density matrices produced by unrestricted calculations using multideterminantal methods (provided that the computational package makes these density matrices available). In contrast with DVMS methods, analyses such as NBO can provide a listing of localized donor-acceptor interactions (and their relative stabilization energies for single determinant wave functions) that underlie arrow-pushing paradigms. This is all to say that, although the DVMS approach is different in the way that it reveals arrow pushing in a localized bond framework, it is not "unprecedented".

Another concern to be addressed is the use of the term radical reactivity. For reactions such as the Diels-Alder reaction, is a single determinant wave function stable with respect to symmetry-breaking, UHF vs RHF, and other constraints? If not this may indicate that the radical-like character of the Diels-Alder reaction is an artifact of the DVMS analysis rather than a true, energy-lowering property of the variational wave function. For the general reader it is important that the authors emphasize that the Diels-Alder takes place on the singlet surface and that the singlet and triplet states are not degenerate, or close to degeneracies, anywhere along the reaction surface (unlike homolysis of H-F).

Response to Comments, Liu et al.

Reviewers' comments:

Reviewer #1 (Remarks to the Author):

In the manuscript, the authors demonstrate that their recently developed Voronoi tiling method is capable of constructing "curly arrow" reaction mechanisms from ab initio wave functions. The physical basis of the widely used "curly arrow" description of reactions has been unclear and disputed. The method of the authors shows convincingly for a simple SN2 reaction, but also for a 4+2 Diels Alder reaction, that movements of electron distributions, resulting in the "curly arrow" description, can be constructed from a Voronoi tiling based on a $|\Psi|^2$ distribution obtained with Metropolis sampling. This work is of general interest for the chemical scientific community. Although the method behind this work was published by the authors a year ago, the application of the method to reaction paths rather than molecules is original. It is particularly interesting that the author's method does not rely on the MO model.

We thank the referee for the considered comments. We are happy that they appreciated the significance of the work, and particular that our method does not rely on a particular description of the electronic wavefunction.

The paper should be published in Nature after some modifications.

- (1) The authors write (p.2 2nd paragraph): [the curly arrow notation...] is [at] odds with MO theory which tends to delocalise electrons in orbitals. This is not quite correct. It is well known that orbitals are not unique and unitary transformations can provide localised MOs (LMO) yielding the same $|\Psi|^2$ and energy as the canonical MOs. LMOs are known to generally yield Lewis structures and thus also "curly arrows" along reaction paths. The authors should discuss similarities and differences of their method and LMO methods such as Knizia and Klein's (ref [10]).

We appreciate that the canonical orbitals can be transformed into localised orbitals which can yield Lewis structures. This is discussed shortly after paragraph 2. We have altered paragraph 2 to be more precise.

The localization of orbitals is performed according to various criteria after the calculation of delocalized MOs by the HF method. This type of localization is only applicable to a single determinant wavefunction and as such cannot describe many reactions or molecules.

Indeed. This is the motivation for the DVMS approach presented here.

- (2) On p. 8, the authors refer to Knizia and Klein's recent paper (the reference [10] is missing), writing that Knizia's localized orbital analysis agrees with the author's results. The advantage of the new method over the well established orbital analysis remains then unclear (except for the easy use of multideterminant wave functions). It seems that Knizia's results do not fully agree with the author's results when comparing the Claisen reaction in Knizia's paper and the similar Diels-Alder reaction in the manuscript. The discussion here should be clarified.

The citation is added at the relevant point. To be clear, we agree with Knizia et al.'s findings with regard to the S_N2 reaction, which we treated with a single determinant wavefunction. This is now emphasised. The advantage of the present method is its ability to deal with any wavefunction at all, not just single determinant ones.

Claisen rearrangement is a different reaction. We do not wish to confuse the readers, but have added:

"While our results for the Diels-Alder reaction support a homolytic mechanism, we do not propose that this necessarily applies to the similarly pericyclic Claisen rearrangement which was found by Knizia and Klein to undergo a cyclic movement of three electron pairs, as their method is bound to conclude."

- (3) The authors sometimes use "sub-tiles" demonstrated in Fig. 2. It should be clarified what criteria are used to split a tile into sub-tiles.

We have added:

"An automated method to determine when to split into sub-tiles can be developed through a clustering analysis. The expectation maximisation algorithm can be adapted to identify whether the cloud of DVMS walkers exhibits significant bimodal character. The expectation maximisation algorithm as a cluster analysis method is well suited to identifying the overlapping clusters that would be expected to occur as wavefunction tiles bifurcate into structures well represented by multiple sub-tiles."

Reviewer #2 (Remarks to the Author):

This is a curious manuscript that uses largely unknown (but interesting) ideas to analyse molecular wave functions. My conclusion is that the authors claim that no other methods can substantiate/illustrate "curly-arrow" (arrow-pushing) interpretations of mechanisms is flawed. One may utilise Natural Bond Orbitals to do this, as is well-known. My feeling is that this flaw suggests that this otherwise interesting work should be sent to another journal (perhaps JACS?) rather than to Nature.

We thank the referee for finding the work interesting and suggesting that it is suitable for JACS (IF=13.9), which is a highly regarded journal.

We do not claim that no other method can do it. We claim that this is the only intuition-free, general method capable of treating a general, “black box” wavefunction, or indeed multiconfigurational wavefunctions.

Reviewer #3 (Remarks to the Author):

This manuscript describes the application of a very interesting "tile-based" analysis of multi electron wave functions to examine localized electron densities and the movement of densities during chemical reactions. The DVMS method has been described separately and is not the focus of the manuscript. Rather the focus of this manuscript is the application of this method to the analysis of electron density flow during chemical reactions for both single determinant and multi-determinant wavefunctions. Although described reasonably clearly, the way the method "works" is beyond the general readership but the way the method "works out" is relatively clear. Overall, this is a novel and intriguing addition to the literature for which publication is warranted, providing that some clarity is provided on the following points.

We thank the referee for the considered comments.

In the analysis of wave functions and densities, uniqueness of a particular representation is a desirable property. It would be helpful if the authors explicitly addressed the following points:

(1) what is the criterion for convergence of the DVMS representations

We have added:

“The convergence of each Monte- Carlo simulation was judged by the observed variation of the position of the Voronoi sites. The certainty of these positions was gauged by the error of the mean of a series of independent Monte-Carlo simulations.”

We run these with different random seeds and starting configurations.

(2) are the qualitative DVMS representations sensitive to the basis set size?

No. We have added: “The qualitative nature of the tiles, of systems tested so-far, have been found to be insensitive to the size of the basis set beyond 6-31G(d).[15]” For water, we found that “The results for the larger 6-311G(d,p) basis set were identical within error.”

(3) For multideterminantal wave functions, does the representation exhibit high sensitivity to, say, the size of the active space?

This depends on the wavefunction. Once the key configurations are included which account for static correlation, the structure will settle down. For C_2 , we found that two major

configurations give rise to the qualitative structure, and that a single configuration give a completely different answer. For benzene (unpublished) we have found that 7 configurations accounts for correlation between the two Kekule structures. In the manuscript, we have added: "However, some wavefunctions require a number of configurations before qualitative convergence."

- (4) How does one determine when a representation requires two or more sub-tiles?

This excellent question has been address above (Referee #1, Q3).

To me, it appears that the properties of other bond localization methods are mischaracterized. For example, density-based methods (ALMO, NBO, NOCV, etc) can certainly analyze the alpha and beta one-electron density matrices produced by unrestricted calculations using multideterminantal methods (provided that the computational package makes these density matrices available).

In contrast with DVMS methods, analyses such as NBO can provide a listing of localized donor-acceptor interactions (and their relative stabilization energies for single determinant wave functions) that underlie arrow-pushing paradigms. This is all to say that, although the DVMS approach is different in the way that it reveals arrow pushing in a localized bond framework, it is not "unprecedented".

Our method for following the reaction path is unprecedented. As such, we choose to retain this word in the abstract.

We have not really "characterized" these bond localization methods anywhere in the manuscript. Yes, any wavefunction can be reduced to a density matrix but that is not the same as analysing the multiconfigurational wavefunction. Nevertheless, we cannot find instances of such schemes to track the evolution of Lewis structures along a reaction path.

We have added: "No attempts to utilise localisation schemes based on the reduced density matrix for multi-determinant wavefunctions to track the evolution of Lewis structures along a reaction path have been published in the literature."

Another concern to be addressed is the use of the term radical reactivity. For reactions such as the Diels-Alder reaction, is a single determinant wave function stable with respect to symmetry-breaking, UHF vs RHF, and other constraints? If not this may indicate that the radical-like character of the Diels-Alder reaction is an artifact of the DVMS analysis rather than a true, energy-lowering property of the variational wave function.

We do not use the term "radical reactivity", however the term "radical mechanism" was included in the caption for Figure 6. This has now been amended to "homolytic mechanism".

The single determinant wavefunction of the D-A transition state is indeed unstable with respect to symmetry-breaking. Allowing optimization with respect to this instability lowers the energy by about 0.12 hartree. The CI wavefunction used here lowers this energy by a further 0.13 hartree. As such, we do not believe that our findings are an artefact.

For the general reader it is important that the authors emphasize that the Diels-Alder takes place on the singlet surface and that the singlet and triplet states are not degenerate, or close to degeneracies, anywhere along the reaction surface (unlike homolysis of H-F).

Added – “This reaction occurs on the singlet surface, and despite the unpairing of electrons during the reaction, the singlet and triplet states are not near-degenerate.”

Reviewer #1 (Remarks to the Author):

The authors have fully addressed the issues raised in the reviews of the first version. The manuscript should be published.

Reviewer #3 (Remarks to the Author):

This review concerns the revised manuscript. The revisions requested by me and the other reviewers are addressed adequately in the new manuscript. At this stage, I support publication in Nature Chemistry.